# Changes in Soil Characteristics, Microbial Metabolic Pathways, TCA Cycle Metabolites and Crop Productivity following Frequent Application of Municipal Solid Waste Compost

**DOI:** 10.3390/plants11223153

**Published:** 2022-11-18

**Authors:** Lord Abbey, Svetlana N. Yurgel, Ojo Alex Asunni, Raphael Ofoe, Josephine Ampofo, Lokanadha Rao Gunupuru, Nivethika Ajeethan

**Affiliations:** 1Department of Plant, Food, and Environmental Sciences, Faculty of Agriculture, Dalhousie University, Halifax, NS B2N 5E3, Canada; 2United States Department of Agriculture, ARS, Grain Legume Genetics and Physiology Research Unit, 24106 N Bunn Road, Prosser, DC 99350-9687, USA; 3Department of Applied Disasters and Emergency Studies, Brandon University, Brandon, MB R7A 6A9, Canada; 4Department of Food Science and Technology, University of California, Davis, CA 95616, USA

**Keywords:** organic amendment, plant metabolites, soil health, environmental health

## Abstract

The benefit sof municipal solid waste (MSW) compost on soil health and plant productivity are well known, but not its long-term effect on soil microbial and plant metabolic pathways. A 5-year study with annual (AN), biennial (BI) and no (C, control) MSW compost application were carried out to determine the effect on soil properties, microbiome function, and plantgrowth and TCA cycle metabolites profile of green beans (*Phaseolus vulgaris*), lettuce (*Latuca sativa*) and beets (*Beta vulgaris*). MSW compost increased soil nutrients and organic matter leading to a significant (*p* < 0.05) increase in AN-soil water-holding capacity followed by BI-soil compared to C-soil. Estimated nitrogen release in the AN-soil was ca. 23% and 146% more than in BI-soil and C-soil, respectively. Approximately 44% of bacterial community due to compost. Deltaproteobacteria, Bacteroidetes Bacteroidia, and Chloroflexi Anaerolineae were overrepresented in compost amended soils compared to C-soil. A strong positive association existed between AN-soil and 18 microbial metabolic pathways out of 205. Crop yield in AN-soil were increased by 6–20% compared to the BI-soil, and by 35–717% compared to the C-soil. Plant tricarboxylic acid cycle metabolites were highly (*p* < 0.001) influenced by compost. Overall, microbiome function and TCA cycle metabolites and crop yield were increased in the AN-soil followed by the BI-soil and markedly less in C-soil. Therefore, MSW compost is a possible solution to increase soil health and plants production in the medium to long term. Future study must investigate rhizosphere metabolic activities.

## 1. Introduction

The aim of the United Nations sustainable development goal #2 is to provide adequate and consistent nutritious food for the estimated 9.73 billion world population by 2050 [1]. Researchers are therefore, exploring different ways to meet this food gap without endangering agroecological systems and environment. One such approach is amendment of agricultural soils with municipal solid waste (MSW) compost to improve soil organic matter content and soil nutrient status, and to sustain soil ecosystem services and biodiversity with the added benefit of increasing food and nutrition security [2,3]. 

Global generation of municipal solid waste (MSW) is estimated at ca. 2 billion Mt per annum, and it is expected to rise to ca. 2.59 billion Mt by 2030 mainly due to increases in urbanization and changes in people’s lifestyles [4,5]. In Canada like many other countries, at least 40% of the MSW from kitchen, yard, restaurants, hotels, and groceries are organic and compostable [6]. MSW compost is a significant source of nutrients, macromolecules, and other compounds essential for plant growth and development [7,8]. Nonetheless, anecdotal evidence suggests that farmers and the public are concerned about the use of MSW compost for food production due to possible contaminations with microbial pathogens like *Salmonella*, and *Escherichia coli* in addition to pharmaceutical residues, plastics, and other harmful physical objects. However, recent trend shows a gradual change in public perception towards acceptability and application of MSW compost. This trend can be attributed to sustained global climate change and deterioration of soil health, scarcity and increase cost of fertilizers, establishment of compost quality standards in many jurisdictions such as in the European Union, Canada and USA, and increase in global research and promotion of MSW compost [9,10,11,12,13].

By nature, compost is complex and comprises humic substances (i.e., fulvic acids, humic acids and humins), non-humic substances (i.e., nutrient elements, macromolecules, and plant growth promoters) and beneficial microbiome that can positively alter many plant structural, physiological, and biochemical functions [12,13,14]. Soil microbiome and function are the basics for the promotion of the stability, productivity, and sustainability of soil health that ensure essential nutrients cycling for optimum plant growth and development [11]. Previous studies demonstrated varied microbial community composition in mature compost including high diversity of nitrogen (N)-fixing, sulphur (S)-oxidizing and nitrifying bacteria, and the biomass of actinobacteria, but a remarkable reduction in Gram-negative bacteria compared to immature compost [15,16,17]. Furthermore, Kelly et al. [18] reported that long-term application of compost increased microbial communities and their functions that culminated in increased nutrient uptake and plant productivity. Besides, MSW compost was also found to increase the accumulation of plant primary metabolites such as organic acids, essential amino acids, and phospholipids in different crop species [19]. Despite these positive observations, the main limitation of MSW compost is the slow and irregular release of N for plant use [10]. Therefore, the appropriate frequency and rate of MSW compost application will be critical to the determination of N availability to plants, and aversion of a decline in crop productivity. 

Many metabolic pathways have been studied in isolation [20], and there is not much literature to link MSW compost effect to plant metabolic pathways. A study by Neugart et al. [21] indicated that food waste compost increased carotenoids concentrations but reduced glucosinolates and phenolics concentrations in pak choi (*Brassica rapa* ssp. *Chinensis*). According to Zhou et al. [22], plants use their root exudates (e.g., oxalate, malate and citrate) as signals to mobilize specific microbial communities to combat disease pathogens, facilitate nutrient acquisition and crosstalk amongst various plant growth regulators. These mechanisms may influence various plant metabolic pathways but understudied. One such major metabolic pathway is the tricarboxylic acid (TCA) cycle, which involves the interconversion of cytosolic glucose, fatty acids and amino acids to acetyl-CoA or other intermediates for mitochondria energy generation [23,24]. Organic acids are the key intermediate metabolites of the TCA cycle, which is a major metabolic pathway for the adenosine triphosphate (ATP) and nicotinamide adenine dinucleotide phosphate (NADPH) syntheses in all organisms [24,25,26]. The precursor of the TCA cycle is pyruvic acid from cytosolic glucose via glycolysis from which acetyl-CoA is formed [27]. So far, literature on if and how compost regulate TCA cycle intermediate metabolites in plants is scarce. 

A recent study by Rosa et al. [28] showed that as the concentration of water extract of compost applied to maize (*Zea mays*) was increased from 0 to 80 mg/L, root exudate of organic acids associated with the TCA cycle metabolites, i.e., oxalic acid, citric acid, malic acid and succinic acid were increased by more than 100%. This might be excess metabolites that were not used in the TCA cycle activities and may suggests that soils amended with compost can influence the former but understudied. Consequently, we postulate that variations in application frequency of MSW compost will differentially alter soil health, microbiome function, and TCA cycle pathway and crop productivity. The objective of the study was to determine how variations in application frequency of MSW compost affect soil health, soil microbiome, and crop productivity and TCA cycle intermediate metabolites. We previously investigated one plant species and did not determine microbiome function and plant metabolic response. In the present study, we were interested to understand the responses of different crop species, so three test crops were used; namely green beans (*Phaseolus vulgaris* cv. Golden Wax) (leguminous crop), lettuce (*Latuca sativa* cv. Grand Rapids) (leafy vegetable) and beets (*Beta vulgaris* cv. Detroit Supreme) (root vegetable). 

## 2. Results and Discussion

### 2.1. Location Climate 

Overall, the climatic conditions in Brandon, MB during the 5-year research were similar except the low minimum mean temperatures in May 2015 (i.e., 1.8 °C) and 2019 (i.e., 0.4 °C) while there was flood and drought conditions in June (i.e., 106 mm) and August 2016 (i.e., 0 mm), respectively (Appendix A). The annual mean temperature ranged from 15–17 °C and the annual mean precipitation ranged from 41 mm to 70 mm. Overall, the differences in climatic conditions did not adversely impact the research over the 5-year period.

### 2.2. Soil Physical Properties

Continuous application of MSW compost for 5 years remarkably altered soil structure and soil function such as increased soil water content and nutrient status and accessibility to plants compared to the control (Table 1) as previously reported by [29]. After Year 5, soil particle and bulk densities were significantly (*p* < 0.05) reduced in the annual plot (AN-soil) seconded by the biennial plot (BI-soil) and then the control plot (C-soil). The high soil organic matter (SOM) content of the AN-soil and the BI-soil enhanced their respective soil structural properties leading to a significant (*p* < 0.001) improvement in soil water retention, i.e., water-holding, water saturation, and field capacities (Table 1) as previously explained by Rawls et al. [30]. All the measured soil water indices were similar for AN-soil and the BI-soil except for water-holding capacity that was increased significantly (*p* < 0.05) in AN-soil by ca. 12% and ca. 27% compared to the BI-soil and the C-soil, respectively. 

### 2.3. Soil Chemical Properties

Total dissolved solids, which is usually used to estimate the proportion of dissolved organic materials including organic matter and salts, was significantly (*p* < 0.001) higher for the AN-soil by more than 208% and 463% compared to that of the BI-soil and the C-soil, respectively (Table 1). Electric conductivity and salinity, which are indicators of soil fertility status, were significantly (*p* < 0.001) high in the AN-soils., i.e., ca. 204% and ca. 211% compared to the BI-soil and ca. 476% and ca. 476% compared to the C-soil, respectively (Table 1). These were expected due to the high SOM in the AN-soil followed by the BI-soil. This is because SOM is a reservoir of soil nutrients [14]. The chemistry of the soil was influenced by continuous and long-term MSW compost application (Figure 1A–H). The SOM in the C-soil increased slightly from Year 1 up to Year 3 before it declined from 2.4% to 1.9% (Figure 1A). There was a sharp increase in AN-soil and BI-soil SOM from Year 1 with a dip in Year 4 before rising again. The dip in Year 4 was due to the late application of compost in Year 3 because of a delay in MSW compost delivery for the study. SOM of the AN-soil at Year 5 was significantly (*p* < 0.01) increased by ca. 33% and ca. 217% compared to those of the BI-soil and the C-soil, respectively. High SOM is associated with high soil organic carbon (SOC) and ultimately, desirable environmental and soil health [31,32]. In 5 years, the AN-soil pH was in general higher (i.e., between lower-upper difference) then BI-soil and C-soil. Soil pH increased from 7.7 in Year 1 to a range between 8.4 (BI-soil) and 8.7 (AN-soil) in Year 3, before declining slightly to an average of 8.15 in Year 5 (Figure 1B). The increase in AN-soil and BI-soil pH could be due to the intrinsically high Na content of the compost, which is supported by the corresponding increases in AN-soil and BI-soil electric conductivity and salinity (Table 1). The increase in pH can also be attributed to the release of hydroxyl ions from the high organic matter AN-soil and BI-soil, which declined after Year 3. This is because MSW compost has high organic matter content with negatively charged sites that can bind or release hydroxyl ions in acidic and basic soils, respectively to buffer soil acidity [33]. There was a slight increase in C-soil pH from Year 4, which can be ascribed to possible base cations naturally associated with Orthic Black Chernozem solum on moderate to strong calcareous, loamy morainal till of limestone, granitic and shale origin (Newdale series) of the experimental site (MAFRD, 2010). This is evident in the highest Ca and Mg levels in C-soil (Figure 1F,G).

The inherent capacity of the soil particles to adsorb cations (i.e., CEC) was not altered by compost application within the first 3 years of the study (Figure 1C). The AN-soil had a higher CEC compared to the BI-soil in Year 5. There was a dip in AN-soil CEC in Year 4, which was not significantly (*p* > 0.05) different from those of BI-soil and C-soil and cannot be readily explained. Cations like K^+^, Na^+^, Ca^2+^ and Mg^2+^ are retained on negatively charged soil components such as organic matter. According to Solly et al. [34], exchangeable Ca contributes the most (i.e., 59–83%) to CEC at pH > 5.5 with a strong positive relationship existing between CEC and SOM. In the present study, the exchangeable Ca increased from 19.41 meq/100 g from Year 1 to 23.67 and 23.99 and 21.85 meq/100 g in the AN-, BI-, and C-soils, respectively, in Year 5 (data not presented). Therefore, the trend of the AN-soil CEC compared to the BI-soil can be ascribed to the pH range (Figure 1B) and its high SOM (Figure 1A) and exchangeable Ca (Table 2). 

Total nitrogen (N) was highly increased in the AN-soil by ca. 149% and ca. 390% more than the BI-soil and the C-soil, respectively (Table 2). Of particular interest was the high estimated nitrogen release (ENR) in the AN-soil followed by the BI-soil (Figure 1D). ENR is a critical index for the estimation of N availability to plants in the next growing season. Typically, nutrients are slowly released from compost due to slow microbial decomposition and mineralization processes. ENR of the C-soil progressively declined while compost application increased ENR, especially in the AN-soil (Figure 1D). The AN-soil ENR was ca. 23% more than that of the BI-soil from Years 3–5; and ca. 69% and 146% more than the C-soil at Years 3–4 and Year 5, respectively. 

The other major plant required nutrient elements, i.e., P, K, Mg and S were significantly (*p* < 0.05) highest in AN-soil followed by BI-soil and the lowest in the C-soil (Table 2). This is expected due to the variations in frequency of MSW compost addition and the resultant differences in soil organic matter content and chemical indices as shown in Table 1. Besides, the trends in percentage P saturation (Psat%) (Figure 1E) and K/Mg ratio (Figure 1F) were similar. That is, AN-soil > BI-soil > C-soil. The Psat% ranged from 4–24%, 4–17% and 4–6% for the AN-soil, BI-soil and C-soil, respectively. Rheault [35] found a threshold range of ca. 6–18% for Manitoba soils with the different types of soil. Therefore, the addition of MSW compost increased Psat% to a maximum (BI) or exceeded the maximum (AN) threshold. Psat% is a function of soil Ca, Fe and Al contents, and an Indicator for environmental risk assessment [36]. The Psat% seemed to level off after Year 3, which suggested less environmental risk, particularly with the BI-soil compared to the AN-soil. This will require further investigation to ensure safe level of soil P for such an AN-soils. Rheault [35] also explained that stabilization of soil P occurs over time leading to a significant reduction in extractable P. We found that exchangeable K and Mg did not significantly (*p* > 0.05) change in the soils after Year 2 (data not presented). Exchangeable K and Mg in the C-soil changed from 0.41–0.37 meq/100 g and 2.76–2.12 meq/100 g in Years 1 and 5, respectively. This can be attributed to the lack of soil amendment and continuous soil nutrient depletion. For the AN- and BI-soils, exchangeable K was increased from 0.41 meq/100 g in Year 1 to 2.77 and 2.09 meq/100 g in Year 2, after which it did not change significantly (*p* > 0.05). Similarly, exchangeable Mg remained the same throughout the study after increasing from 2.76 meq/100 g in Year 1 to 3.78 meq/100 g for the AN-soil and 3.41 meq/100 g for the BI-soil, respectively, in Year 2. These results suggested that MSW compost amendment increased K and Mg availability to plants compared to the control. However, these cations remain the same due to consistency of crop species and MSW compost, i.e., compost type, amount, and time of application. The soil K/Mg ratio was less than one over the entire study period irrespective of the treatment (Figure 1F). This suggested inadequate K for plant use at time of soil sampling. A desirable soil K/Mg ratio is between 2–10. There was a sharp decline in Ca/Mg ratio in Year 2 but rose almost linearly with the highest increase recorded in the AN-soil followed by the BI-soil and then the C-soil (Figure 1G). The high Ca/Mg ratio in AN-soil suggested improved soil structure, and improved porosity and aeration. An exchangeable sodium percentage (ESP) of more than 10 suggests sodic soils. Overall, exchangeable Na was increased in the AN-soil and the BI-soil, which shows that the application of MSW compost increased soil sodicity (Figure 1H), especially in the AN-soil. The general trend for the soil macro-elements P, K, Mg and Ca was AN-soil > BI-soil > C-soil (Table 2). A similar trend was observed for the soil micro-elements sulphur (S), boron (B), iron (Fe), manganese (Mn), molybdenum (Mo), sodium (Na), copper (Cu) and zinc (Zn). However, soil content of cobalt (Co), chromium (Cr) and nickel (Ni) did not change (Table 2), irrespective of the differences in soil treatment. This suggested that soil amendment with this MSW compost will not have any negative environmental impact. 

### 2.4. Soil Microbial Communities

Overall, Basidiomycota was the most relatively abundant fungal phylum found in the microbiome in all the soils, and it was represented by ca. 42% of all ITS reads (Appendix A) and contained 24%, 11% and 6% of the ITS reads annotated as *Tremellomycetes*, *Agaricomycetes*, and *Ustilaginomycetes*, respectively. *Mortierellomycota* were the other most abundant phyla represented by 42% of all the ITS reads. The ITS reads annotated as *Mortierellomycetes*. *Mortierellales*, *Filobasidiales*, *Cystofilobasidiales*, *Agaricales*, and *Ustilaginales* were the most abundant fungal orders represented by 41%, 15%, 9%, 9%, and 5%, respectively. On the other hand, *Actinobacteria*, *Proteobacteria*, *Acidobacteria*, *Chloroflexi*, and *Bacteroidetes* were the most abundant bacterial taxa and were represented by 35%, 23%, 11%, 11% and 5%, respectively (Appendix A). Based on total percentage reads, the most abundant bacterial taxa were *Rhizobiales* (10% of total reads), *Rubrobacterales* (7% of total reads), *Acidobacteria* Subgroup 6 (6% of total reads), *Propionibacteriales* (4% of total reads), and *Gaiellales* (4% of total reads).

In general, compost amendments significantly (*p* < 0.05) influenced soil bacterial community structure and diversity (Figure 2, Appendix A). Unlike bacteria, fungi communities were less affected by the MSW compost application (Figure 2A). 

Approximately 44% of bacterial community variation could be attributed to compost amendment. Moreover, compost significantly (*p* < 0.001) increased the relative number of observed bacterial features and Shannon diversity (Table 3). More specifically, several Actinobacteria classes such as Actinobacteria, Thermoleophilia, Rubrobacteria, Subgroup 6, and MB-A2-108 were less abundant in compost amended soils, i.e., AN- and BI-soils (Figure 2B). The relative abundances of Alphaproteobacterial and Gemmatimonadetes were also reduced by compost application. On the other hand, Deltaproteobacteria, Bacteroidetes Bacteroidia, and Chloroflexi Anaerolineae were overrepresented in compost amended compared to the C-soil (Figure 2B).

The relative abundances of *Alphaproteobacterial* and *Gemmatimonadetes* were also reduced by compost application. On the other hand, *Deltaproteobacteria*, *Bacteroidetes Bacteroidia* and *Chloroflexi Anaerolineae* were overrepresented in compost amended compared to the C-soil (Figure 2B). Our previous study showed that *Bacteroidetes* were among the 10 most abundant taxa in MSW compost sampled across Nova Scotia composting facilities [37]. Therefore, *Bacteroidetes* class *Bacteroidia* (Figure 2B) might be enriched in the MSW compost before adding to the soil resulting in the increase in relative abundance of this class in compost treated soils. The relative abundances of the two most abundant microbial taxa in our previous compost study *Alphaproteobacterial* and *Actinobacteria* [37], were reduced with frequent compost application in the present study (Figure 2B). 

As such, the observed changes in microbial abundances could be due to promotion or repression of soil microbial growth due to differences in compost application frequency. Furthermore, the present results showed that bacterial metabolic pathways were influenced by frequency of compost application, which will be discussed later in this report. Specifically, compost application significantly (*p* < 0.01) increased fungal observed features, but not Shannon diversity (Table 3). Approximately, 9% of variations in fungal community can be attributed to the MSW compost application (Appendix A). We also did not detect any fungal classes differentially represented between treated and untreated soils. This agreed with our previous report that prokaryotes and eukaryotes differ in their responses to environmental factors [38]. 

To understand the effect of compost application on microbiome function, we extrapolated functional profile of bacterial community based on 16S rRNA marker gene using PICRUS2 software. Non-metric multidimensional scaling identified visual differences in functional profiles between bacterial community from soils with different compost application frequency. Additionally, ca. 38% KO and ca. 40% pathway of functional variation was attributed to differences in the frequency of compost application (Table 4). 

When functional profiles of bacterial communities from the AN-soil and the BI-soil were combined and compared to the C-soil, ca. 23% KO and 26% pathway of functional variations were attributed to the former. With reference to C-soil, the BI-soil had lower effect on functional variation., i.e., ca. 15% KO and ca. 18% pathway as compared to those for the AN-soil., i.e., ca. 43% KO and ca. 46% pathway. Interestingly, functional profiles of bacterial communities from the AN-soil and the BI-soil also differed significantly (*p* < 0.001). Obviously, ca. 28% KO and ca. 27% pathway of functional variation was explained by the annual or the biennial compost application compared to the control.

### 2.5. Soil Microbiome Function

In total, 205 pathways were differentially represented in microbiomes from the AN-soil, BI-soil, and the C-soil out of which 18 were overrepresented in the AN-soil and the BI-soil compared to the C-soil (Figure 3). The B-soil had medium level of functional profile of bacteria community compared to the AN-soil (high) and the C-soil (low) on the NMDS plot (Figure 3A). These 18 metabolic pathways were menaquinones (MK) and demethylmenaquinones (DMK) biosynthesis, i.e., superpathway of menaquinol-9 biosynthesis, superpathway of menaquinol-10 biosynthesis; superpathway of menaquinol-6 biosynthesis I, superpathway of demethylmenaquinol-6 biosynthesis I and superpathway of demethylmenaquinol-9 biosynthesis; fatty acid biosynthesis, i.e., oleate biosynthesis IV (anaerobic), stearate biosynthesis II (bacteria and plants), palmitoleate biosynthesis I (from (5Z)-dodec-5-enoate), (5Z)-dodec-5-enoate biosynthesis and superpathway of fatty acid biosynthesis initiation; DNA and RNA structures and enzymes cofactors, i.e., superpathway of purine nucleotides de novo biosynthesis II and pyrimidine deoxyribonucleotides de novo biosynthesis II; energy production, i.e., superpathway of thiamin diphosphate biosynthesis I, NAD biosynthesis II (from tryptophan), superpathway of thiamin diphosphate biosynthesis II, superpathway of pyridoxa–phosphate biosynthesis and salvage and carbon fixation, i.e., reductive acetyl coenzyme A pathway (Figure 3B).

The extent to which plants form mutually beneficial partnership with rhizosphere microbiome is dependent on the genotypic characteristics of the plant as explained by Kelly et al. [18]. This manifested in the plant growth and yield components of the different species in Table 5 and Table 6. According to Zhou et al. [22], plants use their root exudates to mobilize specific microbial communities to combat disease pathogens, facilitate nutrient acquisition and crosstalk amongst various plant growth regulators, and to modulate signaling pathways and increase productivity. This can explain the results of the present study as shown below but will need to be validated in future studies.

### 2.6. Crop Morpho-Physiology 

Application of MSW compost significantly (*p* < 0.05) increased SPAD (soil plant analysis development) value of leaf greenness (Table 5), which can be used to estimate leaf chlorophyll content because of the high positive regression coefficient (R^2^ > 0.93) between the two [39]. F_o_ was comparatively high in all the control but did not differ between AN and BI treated crops. Fm was not significantly (*p* > 0.05) altered by MSW compost irrespective of the crop. F_v_ of lettuce and beets were not altered by compost, but it was significantly (*p* < 0.05) higher in the AN-green beans compared to the BI-green beans and the C-green beans. F_v_, F_v_/F_m_ and F_v_/F_o_ were similarly high (*p* < 0.05) in AN- and BI-green beans and lettuce compared to the control plants (Table 5). Chlorophyll fluorescence indices were not altered in beets except for F_o_ (Table 5). Compost alteration of fundamental plant structural, biochemical, and physiological functions [14] translated into increased leaf chlorophyll content and photosynthetic activities.

Although the MSW compost only increased F_v_/F_m_ and F_v_/F_o_ of green beans grown in soils applied annually with compost, the comparatively low values of these indices in all the control plants suggest stressful conditions (Table 5). It is well established that nutrients imbalance can reduce photosynthetic efficiency and plant metabolism [40,41] as found in plants grown in the control plots. Furthermore, P deficiency reduced F_v_/F_m_ while increasing F_o_ [42] with photosystem II being the most sensitive and vulnerable [43]. The comparatively low C-soil nutrients (Table 2) might have caused the nutrient-deficient control plants to switch to a survival mode by growing extensive root system (i.e., SRL) and biomass by way of increased leaf dry matter content and specific stem density (Table 6). According to Lohmus et al. [44], different plant species adopt different adaptation strategies to sustain and improve nutrition, which may include an increase in above-ground biomass or fine root length. 

The positive impact of compost on crop Productivity is well established [2,12]. Plant heights of the AN-green beans and the AN-beets were significantly (*p* < 0.01) increased by ca. 19% and ca. 17% compared to their BI- counterparts; and by ca. 44% and ca. 46% compared to their C- counterparts, respectively (Table 6). Plant heights for the AN-lettuce and the BI-lettuce were similar (*p* > 0.05) but higher than that of the C-lettuce. Plant leaf area for the AN-green beans, AN-lettuce and AN-beets were increased significantly (*p* < 0.001) by ca. 58%, 15% and 44%, respectively, compared to their BI- counterparts; and by ca. 206%, 63% and 217% compared to their C- counterparts, respectively (Table 6). Compost did not affect leaf dry matter content (LDMC) of the lettuce and beets (Table 6). However, compost significantly (*p* < 0.01) reduced LDMC of the AN-green beans and the BI-green beans by an average of 38% compared to the C-green beans. Specific stem density (SSD) of the plants did not change with compost application (Table 6). However, SSD was reduced in the C-green beans by ca. 74% and increased in C-lettuce and C-beets by ca. 57% and 52%, respectively, compared to the average for their AN- and BI- counterparts.

Specific root length (SRL) was higher in the C-green beans and the C-beets than their AN- and BI- counterparts (Table 6). In contrast, SRL was significantly (*p* < 0.01) reduced in the C-lettuce by ca. 45% compared to the average for the AN-lettuce and the BI-lettuce. There was no significant (*p* > 0.05) difference between the AN- and BI-green bean fresh immature pod yield (Table 6). The fresh pod yield of C-green bean was reduced significant (*p* < 0.01) by ca. 35% compared to the average for the AN- and the BI-green beans. The yield difference between the AN- and the BI-lettuce was ca. 18% (Table 6). The average yield for the AN- and the BI-lettuce was ca. 717% more than the C-lettuce. The yield of the AN-beets was significantly (*p* < 0.0001) increased by ca. 20% compared to the BI-beets; and by ca. 386% compared to the C-beets. 

The results proved that the different crop species responded differently to the MSW compost treatment. Additionally, the results of the plant growth indices demonstrated increased plant growth and productivity with the application of MSW compost compared with the control. According to Cornelissen et al. [45], plants with high LDMC (e.g., control plants) have high physical strength for survival under stress conditions and can be associated with long leaf life-span but may be less productive compared to plants with low LDMC (e.g., plants grown in AN-soil and BI-soil). Similarly, a high SSD demonstrates a dense stem that provides structural strength to the plant and an indication of carbon storage [45], which suggest a switch to survival mode of plants in the C-soil compared to plants that were grown in the AN- and BI-soils. Plants with high SRL develop longer roots per dry mass for water and nutrients uptake [45]. It seemed compost applied lettuce had higher SRL but lower in compost applied green beans and only AN-beets compared to the control. It can be suggested that C-soil had low soil nutrient content and as such, plants develop higher SRL to be able to reach available nutrients and water. Lettuce is shallow-rooted and was probably, stressed in the C-soil to the point where SRL was reduced. 

### 2.7. Elemental Accumulation in Harvested Crops

Chemical elements accumulation in the edible portions of the crops was mostly increased by the annual application of MSW compost (Appendix A), which was previously reported in Abbey et al. [19]. Overall, lettuce accumulated most of the analyzed macro- and micro-elements (i.e., type and quantity) compared to the green beans (intermediate) and beets (lowest). On average, lettuce N (ca. 2100 mg/kg) > green bean N (ca. 967 mg/kg) > beets N (ca. 100 mg/kg); lettuce K (ca. 67,333 mg/kg) > green beans K (ca. 27,850 mg/kg) > beets K (ca. 27,200 mg/kg); lettuce Mg (ca. 5170 mg/kg) > green beans (ca. 2468 mg/kg) > beets (ca. 2183 mg/kg); lettuce Ca (ca. 13,033 mg/kg) > green beans Ca (ca. 4133 mg/kg) > beets (ca. 1890 mg/kg); and lettuce Fe (ca. 294 mg/kg) > green beans Fe (ca. 70 mg/kg) > beets Ca (ca. 39 mg/kg). Continuous application of MSW compost did affect how much micro-elements like Cu, Co and Ni accumulated in the tissues of the different crop species. Clearly, beetroots accumulated the highest Na (i.e., ca. 2530–7830 mg/kg) compared to green beans pod (i.e., ca. 30–40 mg/kg) and lettuce leaf (i.e., ca. 2020–3490 mg/kg). Abbey et al. [19] demonstrated that at a consumption rate of 400 g according to recommendations by WHO [46] and Statistics Canada [47], the crops are safe for human consumption and health based on previous analyses of bioaccumulation factor, estimated daily intake and health hazard quotient.

### 2.8. TCA Cycle Intermediate Metabolites

Plant metabolites involved in the TCA cycle were highly (*p* < 0.001) influenced by MSW compost application frequency (Figure 4A–F). Overall, glucose content was similarly high in lettuce and green beans but low in beets. We found that as part of the survival and adaptation modes, lettuce plants grown in the C-soil accumulated more glucose than their counterparts in the AN-soil and the BI-soil (Figure 4A).

This conforms to previous observation on glucose accumulation in N-deficient plants [26,44]. AN-green beans glucose was ca. 80% and 37% more than that of the BI-green beans and the C-green beans, respectively. The AN-beets and the C-beets glucose were not significantly (*p* > 0.05) different, and their average was over 43% more than that of the BI-beets. The AN-green beans and the BI-green beans had similar and high pyruvic acid content than that of the C-green beans (Figure 4B). The general trend for pyruvic acid was green beans > beets > lettuce. For lettuce and beets citric acid contents, AN = BI > C (Figure 4C). In contrast, citric acid was significantly (*p* < 0.001) increased by ca. 147% and 270% in the BI-green beans compared to the AN-green beans and the C-green beans, respectively.

The AN-lettuce and BI-lettuce α-ketoglutaric acid contents were not significantly (*p* > 0.05) different and was less than that of the C-lettuce (Figure 4D). The C-lettuce α-ketoglutaric acid content was ca. 208% more than the average for the AN-lettuce and the BI-lettuce. There was no significant (*p* > 0.05) difference in α-ketoglutaric acid content between the AN-green beans and the BI-green beans or the AN-beets and the BI-beets. Succinic acid was significantly (*p* < 0.05) increased in the AN-lettuce by ca. 38% compared to the average for the BI-lettuce and the C-lettuce (Figure 4E). Likewise, succinic acid was significantly (*p* < 0.001) increased in the AN-green beans by ca. 103% compared to the average for the BI-green beans and the C-green beans. Succinic acid content was not significantly (*p* > 0.05) different between the BI-lettuce and the C-lettuce or between the BI-green beans and the C-green beans (Figure 4E). Moreover, the AN-lettuce fumaric acid was more than 17% and 63% compared to the BI-lettuce and the C-lettuce, respectively (Figure 4F). The trend for fumaric acid contents in the green beans and the beets was different from that for the lettuce. Fumaric acid content was highest in the C-green beans, and it was ca. 19% and 71% more than those for the AN-green beans and the BI-green beans, respectively. However, the AN-green beans had ca. 44% more fumaric acid than its BI- counterpart. In general, the application of MSW compost reduced fumaric acid content in the beets. This was confirmed by the over 509% increase in the C-beets fumaric acid content compared to the average for the AN-beets and the BI-beets.

The biennially treated plants consistently had high pyruvic acid and citric acid contents (Figure 4). α-ketoglutaric acid is formed from oxidation of isocitrate and found to be similar in plants grown with annual and biennial MSW compost application. Additionally, plant tissue contents of succinic acid (Figure 4E) and fumaric acid (Figure 4F) suggested that much of the high α-ketoglutaric acid in C-lettuce was not converted or its synthesis exceeded the rate at which it was converted to succinic acid. The oxidized succinic acid, i.e., succinate is converted to fumarate and then to malate and oxaloacetate to close the TCA cycle. Fumaric acid was high in AN-lettuce, C-green beans, and C-beets. The interconversion of these intermediate metabolites in the TCA cycle starting from photosynthesis to respiration is tightly regulated by enzymes [23]. The conversion is also dependent on plant species and type of tissue as demonstrated in the present study and in previous report by Fernie and Martinoia [48]. It is obvious from the present study that the composition of these metabolites can vary with plant species, plant developmental stage and type of plant tissue. However, further work will be required to investigate impact on respiratory products such as ATP and NADPH. According to van der Merwe [20], a reduction in TCA cycle activity in a given tissue (e.g., beet roots) reduces respiration and energy generation in that tissue. However, green tissues (e.g., lettuce and green beans) can compensate for TCA cycle deficiencies through photosynthesis and photorespiration as explained by Kromer [49].

### 2.9. Multivariate Assessment

To further examine the relationship among key variables—soil nutrients, microbiome function and citric acid intermediate metabolites—we first constructed a multivariate 2-D PCA biplot using an extrapolated functional profile of bacterial communities based on the 16S rRNA marker gene and the soil nutrient profile (Figure 5).

The two-dimension PCA plot showed that the AN-soil and the BI-soil can be highly and moderately associated with high content of soil elements and the 18 metabolic pathways, respectively, and negatively associated (r = −0.76) with the C-soil (Figure 5A). This is expected because compost increase of soil organic matter, chemical elements and microbiome composition is well established in the literature [2,3,50]. Moreover, DNA and RNA structures and enzymes cofactors (DENOVOPURINE2-PWY and PWY-7187) and MK and DMK pathways had positively strong association (r = 0.8) with soil N and Ca. Several studies confirmed that N source influenced bacteria growth and abundance [51,52,53], which is dependent on their ability to synthesize new cells, i.e., mainly new cellular structures [54]. This suggests that the high N and other elements in the AN-soil (Table 2) met the requirement for bacteria growth more than that of the BI-soil and the C-soil.

The green beans, lettuce and beets tissue N, P, K, Zn, and Cu were positively influenced (r = 0.9) by the 18 microbiome metabolic pathways in the AN-soil (Figure 5B). These pathways showed a moderate influence on citric acid and pyruvic acid accumulation as well as plant height and yield in the AN-green beans and the AN-beets, but less for the AN-lettuce. Under BI-soil, both BI-green beans and BI-beets were associated with high accumulation of pyruvic acid and citric acid, and increased plant height, root length and yield. In contrast, BI-lettuce exhibited high contents of glucose, fumaric acid, α-ketoglutaric acid and succinic acid, which were positively associated (r ≥ 0.78) with all the determined plant tissue elements. On the other hand, BI-lettuce was negatively associated (r ≥ −0.92) with the plant growth components and the metabolic pathways (Figure 5B). The C-lettuce showed a moderate positive association (r = 0.56) with only α-ketoglutaric acid accumulation and root length; and a strong negative association (r = −0.89) with plant tissue elements, plant growth and the remaining TCA cycle intermediate metabolites (Figure 5B). Hence, the low accumulation of metabolites and nutrient elements in edible portions of the C-lettuce, the C-green beans and the C-beets beside the stunted growth and low yield. Overall, MSW compost increased organic acids in the plant tissues that can participate in the TCA cycle pathway for energy generation, especially in plants grown in soils that received compost annually.

A healthy and thriving soil food web is largely dependent on beneficial soil microbial diversity and abundance as influenced by soil organic matter and soil nutrient status [2,55]. Overall, the 2-D PCA biplot showed a strong association between MSW compost enhancement of soil health, particularly plots that received the annual compost application; and high functional microbial metabolic activities that was consistent with previous studies [2,55,56]. Soil microbes utilize carbon and other nutrients in organic matter for their metabolic activities, which facilitate nutrient mineralization and bioavailability [2,56]. Besides, the availability of soil N, P and S are crucial for enhancing microbial amino acids, proteins and nucleotides biosynthetic pathways, and C for structural building and energy production through glycolysis and peptidoglycan biosynthesis. These might have culminated in the high crop performance in the AN-soil followed by the BI-soil and the lowest in the C-soil.

## 3. Materials and Methods

### 3.1. Location and Materials

A 5-year organic field research was performed in Agaard Farms, Brandon, MB, Canada (longitude 99°56′59.9892″ W; latitude 49°50′53.9916″ N; altitude: 409 m above sea level) between fall 2015 and winter 2020. Brandon has dominant to moderate cool, boreal, sub-humid continental climate [57]. The climatic conditions throughout the 5 years of the study were consistent with a 30-year average as presented in Appendix A. The soil of Agaard Farm is Orthic Black Chernozem solum on moderate to strong calcareous, loamy morainal till of limestone, granitic and shale origin under the classification of Newdale series [57]. The City of Brandon waste management facility donated the CQA tested MSW compost, and the 5-year average chemical elements is presented in Appendix A. The seeds for the green beans cv. Golden Wax, lettuce cv. Grand Rapids, and beets cv. Detroit Supreme were purchased from The Green Spot, Brandon, MB.

### 3.2. Field Preparation and Planting

Field preparation, MSW compost application and planting were previously described in Abbey et al. [7,19]. The experimental field measured 80 m × 50 m, and was further divided into three blocks of 20 m × 10 m. Each block was subdivided into three 6 m × 3 m plots, i.e., annual (AN), biennial (BI), and no compost (C, control) plots per block. Separations between blocks was 2 m and plots within blocks was 1 m. MSW compost application rate was ca. 15 t/ha at a bulk density of 650 kg/m^3^. Compost was applied in the fall to AN plot every year or BI plots every two years. Planting begun after the last frost day between 20 and 30 May of each year when the soil temperature rose above 10 °C. Each of the three crops were planted in strips in each plot. The order of planting was rotated each year for five years. Green bean seeds were sown in rows 20 cm apart, and beets and lettuce seeds were sown 10 cm apart. Drip irrigation was applied when the soil moisture content was low or daily in summer. No synthetic chemical fertilizer or pesticide was used. Immature lettuce leaves, beet roots and green bean pods were harvested at edible maturity stage between 45 to 60 days after sowing.

### 3.3. Soil Analysis

Soil samples were randomly collected (n = 5) every year for five years from each plot at the start of the growing season and at peak harvesting time in August using a portable soil auger from 20 cm depth where most of the root mass were located. 300 g of composite soil samples were taken from each treatment plot and the soil physical and chemical properties were analyzed. For microbial analysis, 10 individual soil samples were taken per plot in Year 5. The soils were processed, and DNA was extracted as described in Yurgel et al. [38] and stored at −80 °C for sequencing.

#### 3.3.1. Soil Physical Properties

Soil physical properties and water retention characteristics from the AN, BI and C plots were determined in triplicate in the 5th-year as described by Blake and Hartge [58], Cassel and Nielsen [59], and Danielson and Sutherland [60]. In brief, bulk density (D_b_) was determined from the weight (M) and volume (V_1_) of soil core using a graduated glass cylinder after continuous tapping until there was no observable change in soil volume. For particle density (D_p_), soil lumps were carefully broken, and the volume (V_2_) was recorded using the tapping method.
(1)Bulk density (Db)=MV1
(2)Particle density (Dp)=MV2

Water saturation, field capacity, and wilting point were determined after soil was air-dried under ambient conditions (ca. 22 °C). A known mass of the fresh soil sample (M_s_) was placed in a 15.24 cm plastic pot with drainage holes and weighed (M_sp_). The potted soil was placed in a saucer and saturated with distilled water, and the saturated soil weight (M_sat_) was recorded after 48 h. Then, the saucer was removed for free water to drain out under atmospheric pressure for 72 h and weighed (M_drained_). The drained soil was spread evenly in a flat aluminum tray and air-dried under ambient conditions for 72 h and weighed (M_dried_).
(3)$$\mathrm{Water}\;\mathrm{saturation}\;(S_{c})=\frac{M_{\mathrm{sat}}-{\;M}_{\mathrm{sp}}}{M_{s}}\times 100$$
(4)$$\mathrm{Field}\;\mathrm{capacity}\;(F_{c})=\frac{M_{\mathrm{drained}}-{\;M}_{\mathrm{sp}}}{M_{s}}\times 100$$
(5)$$\mathrm{Wilting}\;\mathrm{point}\;\left(W_{c}\right)=\frac{M_{\mathrm{dried}}-{\;M}_{\mathrm{sp}}}{M_{s}}\times 100$$

Water-holding capacity (WHC) was determined as the difference between field capacity and wilting point., i.e., F_c_ − W_c_.

#### 3.3.2. Soil Chemical Properties

Air-dried soil samples were screened through a 2 mm sieve (Fieldmaster, Guangzhou, China) before sending for elemental analysis at RPC Inorganic Analytical Chemistry Laboratory, Fredericton, NB, Canada every year for five years. A portion of each soil sample was digested according to the United States Environmental Protection Agency (USEPA) method 3050B (Standard Operation Procedures #4.M19). The resulting solutions were analyzed for elemental composition by inductively coupled plasma-mass spectrometry/inductively coupled plasma optical emission spectroscopy (ICP-MS/ICP-ES) using EPA method 200.8/EPA 200.7 (Standard Operation Procedures #4.M01/4.M29). Organic matter content, cation exchange capacity and estimated nitrogen release were also reported. Portions of the soil samples were leached in dilute potassium chlorate for 1 hr, and the leachate was filtered and analyzed calorimetrically for nitrate-nitrite [61]. In brief, soil from each treatment plot was separately added to deionized water at a ratio of 1:4 and stirred vigorously for 2 min. In Year 5, the potential hydrogen concentration (pH), electric conductivity, total dissolved solids, and salinity were determined from the supernatant using ExStik^®^ EC500 instrument (Extech Instruments Corporation, Nashua, NH, USA), and turbidity using an Oakton Turbidimeter (T-100) (Oakton Instruments, Vernon Hills, IL, USA).

#### 3.3.3. Soil Microbial Properties

##### DNA Extraction and Sequencing

In Year 5, DNA extraction was carried out using E.Z.N.A soil DNA isolation kit (Omega Bio-tek, Norcross, GA, USA) according to manufacturer’s protocol and the modified method of Yurgel et al. [38]. DNA quality and concentration were measured using a NanoDrop 1000 spectrophotometer (Thermo Scientific, Waltham, MA, USA). At least 10 µL of DNA samples were sent to Dalhousie University IMR centre (http://imr.bio/; accessed on 5 November 2021) for V6-V8 16S rRNA gene (16S; forward: ACGCGHNRAACCTTACC; reverse: ACGGGCRGTGWGTRCAA), and ITS2 gene (ITS2; forward: GTGAATCATCGAATCTTTGAA; reverse: TCCTCCGCTTATTGATATGC) library preparation and sequencing. Samples were multiplexed using a dual-indexing approach and sequenced using an Illumina MiSeq with paired-end 300 + 300 bp reads. All PCR procedures and Illumina sequencing details were as previously described by Comeau et al. [62]. All sequences generated in this study are available in the NCBI sequence read archive under the accession numbers PRJNA883731 (16S rRNA) and PRJNA883747 (ITS).

##### Amplicon Variant Sequence (AVS) Picking and Statistical Analyses

A Microbiome Helper standard operating procedure [63] and QIIME2 wrapper scripts [64] were used to process and analyze the sequencing data. Briefly, the primers were trimmed and overlapping paired-end reads were stitched together followed by quality filtering and open-reference AVSs picking. We filtered out AVSs that contained fewer than 0.1% of the total sequences to compensate for MiSeq run-to-run bleed-through. The complete statistics of Illumina sequencing data analysis is presented in Appendix A. In brief, 892,582 and 188,044 high-quality non-chimeric reads were obtained from 16S rRNA and fungal internal transcribed spacer 2 (ITS2). These sequences were clustered into 55,483 (16S rRNA) and 1281 (ITS2) AVS. AVSs annotated as mitochondria and chloroplast were removed and the datasets were normalized to the depth of 6350 and 1186 frequencies resulting in the normalized datasets comprising 1768 (16S rRNA) and 721 (ITS2) AVSs, respectively.

##### Microbiome Functional Analysis

To understand the impact of MSW compost on microbiome function, we extrapolated functional profile of bacterial community based on 16S rRNA marker gene. Functional potentials of the bacterial community were predicted using the AVS tables and reference sequences generated by QIIME2, which were processed by PICRUSt2 software [65]. Abundance tables were generated both for complete MetaCyc functional pathways as well as individual enzymes categorized by KEGG Orthology (KO) numbers.

##### Bioinformatic Analysis of Amplicon Sequencing and PICRUSt2 Outputs

Alpha-diversity, i.e., Chao1 richness, Simpson evenness and Shannon diversity, and beta-diversity metrics were generated using QIIME2 [66]. Variations in sample groupings explained by beta-diversity distances (i.e., Adonis tests, 999 permutations) were run in QIIME2 to calculate how sample groupings are related to microbial community structure and function. Differential abundances bacterial taxa and pathways were determined using ALDEx2 with Benjamini-Hochberg corrected *p* value of Welch’s *t*-test (*p* < 0.01) [67]. Non-metric multi-dimensional scaling (NMDS) plots were developed based on Bray–Curtis distances using Vegan R package [68]. Heatmap and NMDS plots were build used gglpot2 R package [69].

### 3.4. Plant Morpho-Physiology and Yield

#### 3.4.1. Leaf Pigmentation and Plant Photosynthetic Efficiency

Leaf chlorophyll and anthocyanin contents were estimated from four leaves per plant using SPAD 502DL Plus Chlorophyll Meter (Spectrum Technologies Inc., Aurora, IL, USA) and Anthocyanin content meter ACM-200 plus (Opti-Science Inc., Hudson, NH, USA), respectively. A portable OS30p_+_ Chlorophyll fluorometer (Opti-Science Inc., Hudson, NH, USA) was used to record leaf fluorescence indices between 08:00 and 11:00 am Central Standard Time (GMT-6). Briefly, to measure fluorescence indices, the middle portion excluding mid-vein of each selected leaf was clipped with light exclusion clips for 20 min prior to recording minimum (F_o_), maximum (F_m_), variable (F_v_) fluorescence indices, potential photosynthetic capacity (F_v_/F_o_) and maximum quantum yield of photosystem II (F_v_/F_m_). All of these plant pigmentation and photosynthetic efficiency data were collected at eight weeks after planting in Year 5.

#### 3.4.2. Plant Growth and Yield

Plant growth and yield components were taken from six plants in the middle of each plant row per species per plot in Year 5. Data collected were plant height using a 60 cm ruler; leaf area (A) was measured using LI-3000C portable leaf area meter (LI-COR Biosciences, Lincoln, NE, USA); and leaf dry matter content (LDMC) was determined from the ratio of oven-dry mass (M_d_) to a water-saturated leaf mass (M_f_). In brief, leaf samples were submerged in 250 mL distilled water for 9 h to obtain M_f_ before oven-dried at 65 °C for 48 h in a mechanical convection oven (Cole-Parmer Instrumental Co., Vernon Hills, IL, USA) to obtain LDMC.
(6)$$\mathrm{LDMC}=\frac{M_{d}}{M_{f}}$$

For specific stem density (SSD), diameter (d) of a 10 cm long (L) stem from plant with full green foliage was measured using a pair of calipers and the volume (V) of the stem was calculated using the oven-dry mass of the stem (M).
(7)$$\mathrm{SSD}=\frac{M}{V};V={\left(0.5d\right)}^{2}\times \pi \times L$$

Specific root length (SRL) was determined from plant absorptive roots that were excavated gently from the soil using a spade and taking extra care not to lose more that 5% of the root mass, and to ensure that the total root length was intact before washing under running water until clean. A 30 cm ruler was used to measure the root length (R_L_) and the air-dry mass (R_m_) was recorded after drying under ambient laboratory conditions at ca. 22 °C.
(8)$$\mathrm{SRL}=\frac{R_{L}}{R_{m}}$$

Total fresh weight of the edible portions of each vegetable plant species was used to estimate crop yield from the green leaves of lettuce cv. Grand Rapids, roots of beets cv. Detroit Supreme, and immature pods of green bean cv. Golden Wax.

### 3.5. Plant Chemical Composition

Part of the elemental composition data were previously reported in Abbey et al. [7,19]. Briefly, edible portions of the lettuce, beets, and green beans were harvested and oven-dried at 65 °C for 48 h. The dried plant tissues were ground using a hammer mill and screened through a 53-µm sieve after which samples were analyzed as described below.

#### 3.5.1. Plant Elemental Composition

A 30 g sample of each ground tissue was shipped on ice to Research and Productivity Council (RPC) Laboratory, New Brunswick, Canada for further preparation and analysis in Year 5. Portions of the samples were prepared by microwave-assisted digestion in nitric acid (SOP 4.M26). The resulting solutions were analyzed for chemical elements by ICP-MS/ICP-ES (SOP 4.M01). Kjeldahl nitrogen and nitrates and nitrites concentrations were analyzed using the digestion, phenate colourimetry (SOP IAS-M16) and hydrazine red, derivatization, colourimetry (SOP IAS-M48) methods, respectively.

#### 3.5.2. TCA cycle Intermediate Metabolites

A 35 g sample of each ground plant tissue was put in a screw cap tube before shipping on dry ice to Canada’s national metabolomics core facility at The Metabolomics Innovation Centre (TMIC), AB for analysis of targeted organic acids (i.e., α-ketoglutaric acid, succinic acid, fumaric acid, citric acid, and pyruvic acid). The samples were analyzed by the method described in Abbey et al. [19]. Briefly, 150 µL of ice-cold methanol and 10 µL of isotope-labeled internal standard mixture were added to 50 µL of plant extract for overnight protein precipitation. The mixture was centrifuged at 13,000× *g* for 20 min. Afterwards, 50 µL of the supernatant was loaded into 96-deep well plate followed by the addition of 3-nitrophenylhydrazine (NPH) reagent. After incubation for 2 h, BHT stabilizer and water were added before LC-MS injection. Mass spectrometric analysis was performed on an ABSciex 4000 Qtrap^®^ tandem mass spectrometry instrument (Applied Biosystems/MDS Analytical Technologies, CA, USA) equipped with an Agilent 1260 series ultra-high performance liquid chromatography (UHPLC) system (Agilent Technologies, Santa Clara, CA, USA). The samples were delivered to mass spectrometer by LC method followed by direct injection (DI) method.

### 3.6. Experimental Design and Data Analysis

#### Soil Physiochemical Properties and Plant Growth Components

The AN, BI and C treatments and the three crop species; namely, green beans, lettuce and beets were arranged in a randomized complete block design with three replications. The three plant species were treated independently per block. Ten samples of soil were collected per plot and bulked before taking a composite sample. Soil chemical analyses were performed from the composite sample in triplicate. For plants, tissue samples were collected from 10 plants in the middle of the plant row and tissue chemical analyses were performed in duplicate, and the average was presented due to large sample size and limited resources. Data collected on soil physical and chemical properties and crop growth were subjected to one-way analysis of variance (ANOVA) using Proc Mixed in SAS version 9.4 (SAS Institute Inc., Cary, NC, USA). Fisher’s least significant difference (LSD) test was used for mean separation at α = 5% when ANOVA showed significant difference at *p* ≤ 0.05. Data analysis of the metabolites was done using Analyst 1.6.2. and XLSTAT version 19.1 (Addinsoft, New York, NY, USA). Microsoft Excel was used to plot graphs.

## 4. Conclusions

The present study focuses on application frequency of MSW compost effect on soil health, soil microbiome function, and crop productivity and TCA cycle intermediate metabolites. There are clear indications of positive impact of annual application of MSW compost on plant tissue nutrient status, photosynthetic activities, and primary metabolites accumulation. Currently, farmers are faced with many challenges related to soil health, which is exacerbated by sustained global climate change, and the current global shortage and high cost of fertilizers. Therefore, the present work is urgently needed and timely. We concluded that MSW compost application is safe and effective for food production. We affirm that annual application of MSW compost promoted positive association between microbiome function and ecosystem services for the benefit of crop plants. Therefore, annual application of MSW compost can be classified as climate-smart and sustainable practice for the maintenance or rejuvenation of agricultural soils for nutrient-dense food production over the medium to long-term. This will benefit farmers and if implemented, will immensely benefit consumers with safe, nutritious and affordable food crops. Plants are selective in the composition of rhizosphere environment and microbial communities. As such, our next step is to investigate the rhizosphere composition of different plant species in compost amended soil.

## Figures and Tables

**Figure 1 plants-11-03153-f001:**
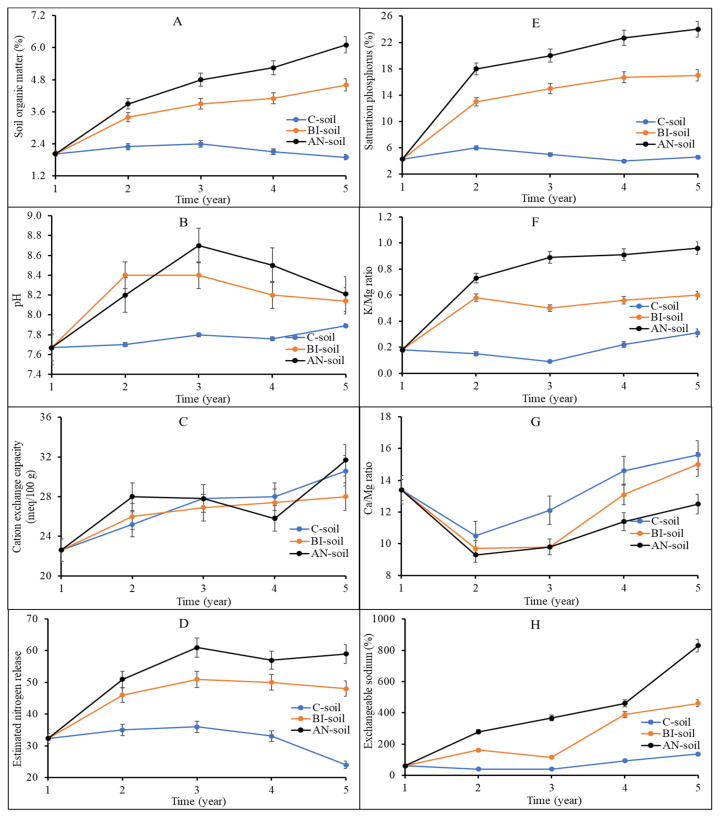
Changes in soil fertility status over five years of application of municipal solid waste compost at varying application frequency. C-soil, BI-soil and AN-soil represent no compost (control), biennial and annual application of compost, respectively. (**A**–**H**) is panel label. Vertical bars represent standard error bars (N = 9). K/Mg, potassium to magnesium ratio and Ca/Mg, calcium to magnesium ratio.

**Figure 2 plants-11-03153-f002:**
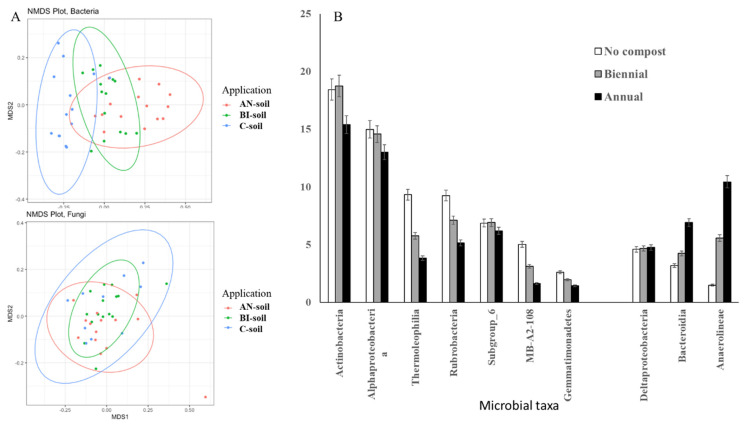
Diversity and structure of soils under different rates of compost application. (**A**): Non-metric multidimensional scaling plot of communities in soils under different rates of compost application. The analysis is based on Bray–Curtis distances of 16S rRNA (**top**) and ITS2 (**bottom**) samples. (**B**): Microbial taxa that were significantly overrepresented in comparison between soils with different rates of compost application. AN, BI, and C are annual, biennial and no compost (control) application, respectively. Corrected *p*-values (*q*-values) were calculated based on Benjamini-Hochberg FDR multiple test correction. Features with (Welc’s *t*-test) *q* value < 0.01 were considered significant and were retained. Only microbial taxa represented by >2% total reads were shown at the class level.

**Figure 3 plants-11-03153-f003:**
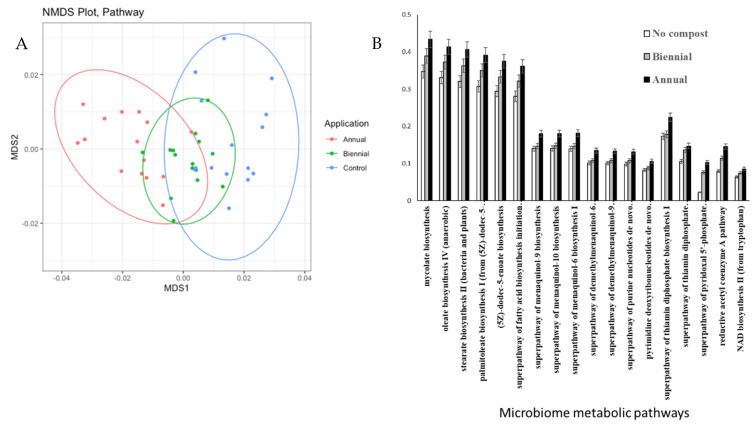
Functional profile of bacterial community based on 16S rRNA marker gene. (**A**)–Non-metric multidimensional scaling plot of community functions in soils under different rates of compost application. The analysis is based on Bray–Curtis distances of functional composition of bacterial community evaluated using PICRUSt. (**B**)–Pathways those were significantly overrepresented in Biennially and Annually treated soil compared to the untreated soil. Corrected *p*-values (*q*-values) were calculated based on Benjamini–Hochberg FDR multiple test correction. Features with (Welch’s *t*-test) *q*-value <0.01 were considered significant and were thus retained.

**Figure 4 plants-11-03153-f004:**
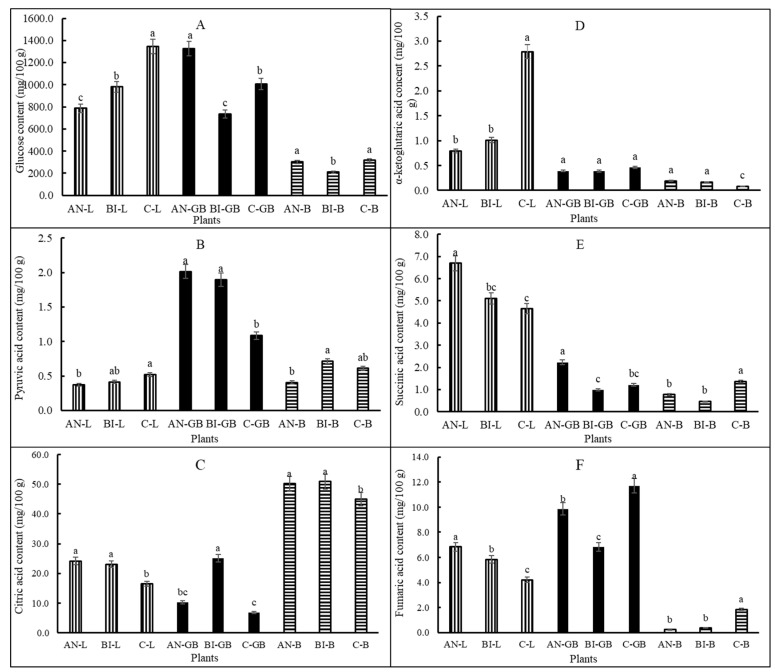
Plant metabolites of the tricarboxylic acid cycle as influenced by application of municipal solid waste compost at varying application frequency. Bars with vertical lines represent lettuce, solid black bars represent green beans and bars with horizontal lines represent beets; C-L, C-GB and C-B represent lettuce, green bean and beets grown with no compost (control), respectively; BI-L, BI-GB and BI-B represent lettuce, green bean and beets grown with biennial compost application, respectively; and AN-L, AN-GB and AN-B represent lettuce, green bean and beets grown with annual compost application, respectively; (**A**–**F**) is panel label. Differences in the letters on bars indicate significant differences at *p* < 0.05 (N = 9).

**Figure 5 plants-11-03153-f005:**
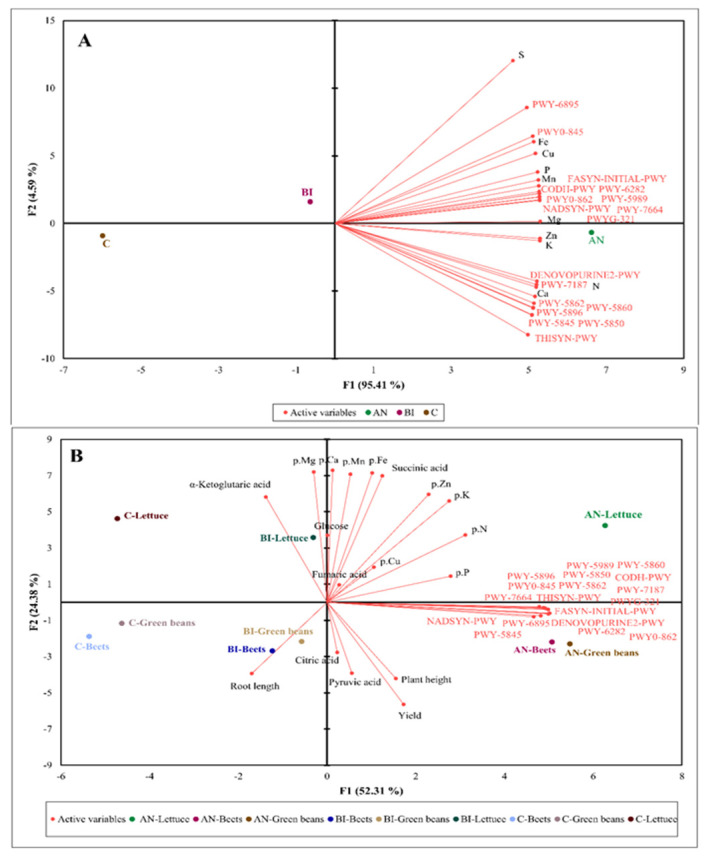
Multivariate analysis using principal component biplot. (**A**): shows association amongst municipal solid waste compost application frequency, soil elements and soil microbial function. (**B**): shows association amongst crop species, growth and yield components, plant tissue elements, tricarboxylic acid cycle metabolites and soil microbial function. AN, BI, and C are annual, biennial and no compost application, respectively. S, sulphur; Fe, iron; Cu, copper; P, phosphorus; Mn, manganese; Ca, calcium; N, nitrogen; Zn, zinc; K, potassium; Mg, magnesium. p represent plant tissue.

**Table 1 plants-11-03153-t001:** Soil physical and chemical properties of experimental plots in Year 5.

Soil Properties	Compost Application		
	Annual	Biennial	Control
Saturation capacity (%)	49.33 ± 1.23 a	46.39 ± 1.91 a	39.65 ± 0.98 b
Field capacity (%)	41.24 ± 0.98 a	35.37 ± 1.00 ab	30.49 ± 1.12 b
Wilting capacity (%)	6.40 ± 1.09 a	4.69 ± 0.09 ab	4.16 ± 1.11 b
Water-holding capacity (%)	34.80 ± 2.42 a	30.68 ± 1.49 b	25.33 ± 2.01 c
Bulk density (g/cm^3^)	0.99 ± 0.00 c	1.06 ± 0.01 b	1.22 ± 0.04 a
Particle density (g/cm^3^)	1.22 ± 0.10 c	1.33 ± 0.09 b	1.41 ± 0.12 a
Turbidity (NTU)	584.33 ± 8.14 a	516.22 ± 7.80 ab	453.56 ± 6.77 c
Organic matter content (%)	6.11 ± 1.00 a	4.61 ± 1.23 b	1.93 ± 0.21 c
Total Dissolved Solids (mg/L)	1535.2 ± 10.1 a	498.1 ± 9.91 b	272.4 ± 8.71 b
Electric conductivity (µS/cm)	2175.0 ± 10.8 a	715.2 ± 12.1 b	377.5 ± 10.00 b
Salinity (mg/L)	1040.6 ± 9.90 a	334.8 ± 7.61 b	180.8 ± 6.97 b

NTU, Nephelometric turbidity unit; means sharing the same alphabetical letters within the same row are not significantly different at the 5% level.

**Table 2 plants-11-03153-t002:** Five-year accumulation of essential and beneficial chemical elements in soils as affected by frequency of application of municipal solid waste compost.

Compost Application	Soil Chemical Elements (mg/kg)							
	Total N	Phosphorus	Potassium	Magnesium	Calcium	Sulphur	Boron	Iron
Annual	50.0 ± 3.2 a	274.0 ± 4.1 a	3610.2 ± 6.1 a	5920.1 ± 23.1 a	22,000.8 ± 21.1 a	29.7 ± 4.5 a	13.3 ± 1.2 a	9330.4 ± 9.9 a
Biennial	20.1 ± 2.1 b	195.7 ± 2.8 b	1950.0 ± 6.5 b	4380.1 ± 22.9 b	17,401.3 ± 19.0 bc	28.0 ± 3.9 a	10.0 ± 1.8 ab	8740.6 ± 11.2 b
Control	10.2 ± 1.0 c	95.3 ± 2.3 c	940.4 ± 4.3 c	3220.0 ± 16.1 c	16,100.2 ± 19.0 c	12.7 ± 1.7 b	6.7 ± 1.1 b	7670.0 ± 8.7 c
Compost application	Soil chemical elements (mg/kg)							
	Manganese	Molybdenum	Cobalt	Sodium	Chromium	Copper	Nickel	Zinc
Annual	605.3 ± 8.7 a	0.8 ± 0.0 a	3.4 ± 0.0 a	119.3 ± 2.4 a	9.3 ± 0.0 a	13.3 ± 1.5 a	9.2 ± 0.0 a	59.1 ± 3.1 a
Biennial	511.1 ± 8.7 ab	0.51 ± 0.0 ab	3.1 ± 0.0 a	95.0 ± 1.1 a	8.1 ± 0.4 a	9.7 ± 1.3 a	9.1 ± 0.0 a	40.0 ± 2.3 b
Control	400.4 ± 7.9 b	0.24 ± 0.1 b	2.9 ± 0.2 a	22.3 ± 1.6 b	6.3 ± 0.09 a	4.0 ± 0.2 b	8.2 ± 0.3 a	28.2 ± 1.7 c

Means sharing the same alphabetical letters within the same column are not significantly different at the 5% level.

**Table 3 plants-11-03153-t003:** Microbial alpha-diversity as affected by frequency of application of municipal solid waste compost.

Treatment/Indexes	Observed Features	Evenness	Shannon Diversity
16S			
Control	1026 ± 6.1 b	0.892 ± 0.1 a	8.825 ± 1.2 b
Biennial compost application	1127 ± 5.1 a	0.902 ± 0.1 a	9.139 ± 1.1 a
Annual compost application	1138 ± 5.0 a	0.889 ± 0.1 a	9.023 ± 1.2 ab
ITS			
Control	43 ± 1.3 b	0.793 ± 0.2 a	4.280 ± 0.1 a
Biennial compost application	76 ± 2.2 a	0.734 ± 0.1 a	4.574 ± 0.2 a
Annual compost application	67 ± 1.8 a	0.753 ± 0.1 a	4.521 ± 0.2 a

For each variable, data followed by different letters are significantly different according to Kruskal–Wallis pairwise comparison (*q* < 0.05).

**Table 4 plants-11-03153-t004:** Variation in groupings of functions prediction by PICRUSt2 explained by Bray–Curtis distances as affected by frequency of application of municipal solid waste compost.

Grouping (Subset)	KO	Pathways
Treatments	0.383 ± 0.00 ***	0.400 ± 0.01 ***
Control vs. Biennial and Annual compost applications	0.226 ± 0.00 ***	0.264 ± 0.01 ***
Control vs. Biennial compost application	0.151 ± 0.00 ***	0.182 ± 0.00 ***
Control vs. Annual compost application	0.434 ± 0.01 ***	0.460 ± 0.02 ***
Annual vs. Biennial compost application	0.283 ± 0.00 ***	0.266 ± 0.00 ***

Adonis tests were used to assess whether beta-diversity is related to sample groupings, 999 permutations, R2, *** *p* < 0.001.

**Table 5 plants-11-03153-t005:** Leaf pigmentation and plant photosynthetic efficiency indices as affected by frequency of application of municipal solid waste compost.

Crop	Compost Application	SPAD ^1^ Value	ACI ^2^	Chlorophyll Fluorescence Index				
				F_o_ ^3^	F_v_ ^4^	F_m_ ^5^	F_v_/F_m_ ^6^	F_v_/F_o_ ^7^
Beans	Annual	42.9 ± 1.2 ab	7.34 ± 0.0 a	151.73 ± 3.2 b	407.37 ± 2.9 ab	559.33 ± 3.2 a	0.72 ± 0.0 a	2.82 ± 0.0 a
	Biennial	46.4 ± 1.0 a	7.97 ± 0.2 a	188.47 ± 3.1 b	491.10 ± 1.9 a	682.23 ± 2.8 a	0.72 ± 0.1 a	2.93 ± 0.2 a
	Control	36.3 ± 0.8 c	6.12 ± 0.0 a	225.13 ± 4.4 a	349.20 ± 3.2 b	570.33 ± 2.2 a	0.59 ± 0.0 b	1.72 ± 0.2 b
Lettuce	Annual	42.9 ± 1.2 a	6.32 ± 0.2 a	159.00 ± 3.6 b	574.93 ± 2.1 a	733.93 ± 2.1 a	0.78 ± 0.4 a	3.73 ± 0.1 ab
	Biennial	36.7 ± 0.9 b	5.44 ± 0.3 a	142.70 ± 2.3 b	558.23 ± 2.2 a	706.60 ± 2.3 a	0.79 ± 0.1 ab	3.96 ± 0.0 a
	Control	27.8 ± 1.0 c	3.53 ± 0.3 b	171.67 ± 1.9 a	528.30 ± 2.0 a	700.97 ± 3.1 a	0.75 ± 0.7 b	3.13 ± 0.3 b
Beet	Annual	50.2 ± 1.8 a	7.95 ± 0.0 b	139.30 ± 0.1 b	435.10 ± 2.0 a	573.73 ± 1.0 a	0.75 ± 0.7 a	3.16 ± 0.1 a
	Biennial	38.5 ± 1.0 b	9.45 ± 0.2 b	134.43 ± 1.9 b	444.47 ± 2.3 a	578.70 ± 1.7 a	0.75 ± 0.7 a	3.42 ± 0.1 a
	Control	30.1 ± 0.4 c	18.97 ± 0.4 a	177.13 ± 2.2 a	483.37 ± 2.1 a	660.53 ± 0.9 a	0.72 ± 0.6 a	2.85 ± 0.0 a

^1^ SPAD, soil plant analysis development; ^2^ ACI, anthocyanin content index; ^3^ F_o_, minimum, ^4^ F_m_, maximum and ^5^ F_v_, variable chlorophyll fluorescence indices; ^6^ F_v_/F_m_, maximum quantum yield of photosystem II; and ^7^ F_v_/F_o_, potential photosynthetic capacity. means sharing the same alphabetical letters within the same column are not significantly different at the 5% level.

**Table 6 plants-11-03153-t006:** Plant growth components and yield of green beans (*Phaseolus vulgaris* cv. Golden Wax), beets (*Beta vulgaris* cv. Detroit Supreme) and lettuce (*Latuca sativa* cv. Grand Rapids) as affected by frequency of application of municipal solid waste compost.

Plant Growth Component			Compost Application		
			Annual	Biennial	Control
Shoot	Plant height (cm)	Green beans	79.84 ± 3.2 a	67.33 ± 2.3 b	55.6 ± 1.5 c
		Lettuce	30.10 ± 1.6 a	32.00 ± 1.7 a	24.16 ± 1.3 b
		Beets	40.80 ± 1.4 a	34.98 ± 1.6 b	28.05 ± 1.4 c
Leaf	Area (mm^2^)	Green beans	8525 ± 13.9 a	5398 ± 12.2 b	2782 ± 9.9 c
		Lettuce	23,484 ± 10.7 a	20,442 ± 9.9 b	14,389 ± 8.3 c
		Beets	20,521 ± 20.0 a	14,219 ± 19.0 b	6481 ± 10.8 c
	LDMC (mg/g)	Green beans	121.98 ± 2.2 b	155.70 ± 2.6 b	223.24 ± 2.9 a
		Lettuce	61.06 ± 2.1 a	71.62 ± 2.4 a	75.17 ± 1.9 a
		Beets	137.92 ± 2.9 a	122.08 ± 1.6 a	140.52 ± 1.7 a
Stem	SSD (mg/mm^3^)	Green beans	0.12 ± 0.0 a	0.30 ± 0.1 a	0.08 ± 0.0 b
		Lettuce	0.07 ± 0.3 b	0.07 ± 0.2 b	0.11 ± 0.0 a
		Beets	0.12 ± 0.0 b	0.13 ± 0.0 b	0.19 ± 0.0 a
Root	SRL (mg/g)	Green beans	0.12 ± 0.2 b	0.12 ± 0.0 b	0.27 ± 0.1 a
		Lettuce	0.15 ± 0.0 a	0.14 ± 0.0 a	0.08 ± 0.0 b
		Beets	0.36 ± 0.1 b	0.76 ± 0.1 a	0.77 ± 0.2 a
Edible portion	Yield (kg/m^2^)	Green beans	434.16 ± 4.3 a	411.42 ± 4.2 ab	321.31 ± 8.1 c
		Lettuce	85.00 ± 2.7 a	72.09 ± 1.4 a	10.40 ± 1.3 c
		Beets	311.60 ± 3.0 a	260.00 ± 3.8 b	64.08 ± 3.3 c

LDMC, leaf dry matter content; SSD, specific stem density; SRL, specific root length; means sharing the same alphabetical letters within the same row are not significantly different at the 5% level.

## Data Availability

Not applicable.

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
