# Peer review of "Changes in Soil Characteristics, Microbial Metabolic Pathways, TCA Cycle Metabolites and Crop Productivity following Frequent Application of Municipal Solid Waste Compost"

_plants, 2022, doi:10.3390/plants11223153_

Round 1

Reviewer 1 Report

The current study is on a topic of relevance and general interest to the readers and journals. This article talks about a the effect of compost addition on soil fertility, microbial community and plant physiology ad productivity in long term experiment (5 years) with different compost application management, this topic have great interest to the community and it is in line with the aims of the journal. Although the manuscript is written with a proper English, I found that the authors have to improve it and integrate some part.

Rows 18-20: this sentence looks confused, it is useful use the passive form in this section to better understand the aims of your research, e.s. A 5-year study with annual (AN), biennial (BI) and no (C, control) MSW compost application were carried out and MSW compost effect on soil properties, microbiome function, and the growth and metabolic performances of green bean (Phaseolus vulgaris), lettuce (Latuca sativa) and beets (Beta vulgaris) were assessed.”

Row 21: which elements? Macronutrients, micronutrients? Or in general plant nutrients or others? Please specify

Row 22: if in C soil you have not added MSW compost how can the compost has minor effects on soil fertility? Maybe you have to compare your data with the control soil.

Rows 24-25: Was these microbial groups in all the treatments? Or not specify if your treatments had impacts on microbial composition.

Rows 27-28 Why did you compared the AN treatment with the BI treatment you had to compare both the treatments with the Control.  Because is the control which give you your baseline and then you can establish which is the best treatment.

Row 29: “Overall, the crops perform best in the AN-soil followed by the BI-soil.” Did you compare it with the control? I understand that you have few words for your abstract, but this sentence is uncomplete. Finally, you could add some consideration like MSW compost is a possible solution to increase the soil health and plants production in the long-medium term.

Rows 41-43 Check the formatting.

Rows 60-63: Which is the subject of your comparison? The increases and the reductions must be focused on a baseline, so if you are talking about the raw manure and the final compost you have to explicit it or change the terms increase and reduction maybe with high contents or low content without comparisons.

Row 78: maybe but it is understudied could sound better.

Rows 109-111: It is not clear which is your baseline, if it is the control or is the soil of 5 years ago before the treatments, please if the baseline is the old soil add its features to the table because it is impossible to the readers understand it.

Rows 114-116: I understand that repeat several times AN, BI and C is not the best, but it hard to the readers follow your discussion with former and latter. You can find synonyms for your treatments to change your discussion.

Row 118: Is there a reason why you do not put the standard deviation/standard error on your data? If not add them because is useful for readers to understand the real variability of your samples.

Rows 122-128: Check the formatting.

Rows 112-129: Are you sure that the “mean values” (change also this) are different within the column? Reading the table for me the letters have to be read within the row. Check what you meant.

Rows 146-153: Why did you not described the C soil? At Year 5 there was a pH increase why? In this part you did not report any comparison you could add a sentence like “in 5 years the AN soil pH was in general higher (i.e. between lower-upper difference) then BI and C soils.

Rows 161-162: This sentence is wrong in the figure 1C there are not statistically differences among your treatments and AN soil at year 4 is not the higher. How do you explain the increase of CEC in C soil the fluctuations in AN-soil? Please rewrite the CEC part.

Rows 161-167: Why did you not show the data about exchangeable Ca? Total Ca is no exchangeable please check this.

Rows 181-190: Why did you chose to analyze only the P saturation? In my opinion for the plant nutrition is more important the available P. I agree that P saturation could be useful to prevent eutrophication of water bodies but here you are talking about plant nutrition and soil fertility not only environmental issues.

Rows 190-195: Why did you not talk about exchangeable K, Ca, Mg? The ratio between these elements could be important but are total elements that means that could be not available for plant nutrition.

Rows 201-202: This is very important from the environmental point of view. The fact that these potentially toxic elements (PTEs) are not increased means that adding this MSW compost you are not polluting the soil with PTEs. Please add some consideration in this sense.

Rows 205-2016: Which are the soil that were referred to these data? Please explicit if you are talking about C soil or if it is a general consideration valid for all your treatments.

Row 220: The figure 2 is a bit small, but maybe is due to the formatting of the reviewer version. However, check it. For the B graph there are no error bars or letters to evaluate the statistical differences, please add them maybe both letter and error bars.

Rows 272-274: Here your baseline is C soil, this is different from the rest of the manuscript. In my opinion this is the best way to show your data, evaluate if it is possible change all the manuscript in this way. However, it a mistake that could confuse readers change the baseline in the different sections.

Rows 283-284: BI-soil correct. Again, you changed the baseline chose only one treatment as baseline. Do it in all the manuscript please.

Row 297: The figure 3 is a bit small, but maybe is due to the formatting of the reviewer version. However, check it. For the B graph there are no error bars or letters to evaluate the statistical differences, please add them maybe both letter and error bars.

Row 316: Fm correct

Row 327: Why the F factors are reported with different labels compared with those in the text. Please uniform labels in table and text.

Row 360: Please check the statistics if are different by rows or columns.

Rows 344-371: These results have been not discussed. Please do it.

Row 386-388: the discussion is very poor. Please enhance your consideration maybe highlighting that these foods could be useful for human health.

Row 392-394: Your comment is not general is valid only for lettuce, please specify and describe carefully your results.

Row 395: The figure 4 is low in quality and looks squashed. You should add the legend that allows us to understand that the different textures represent the different crops. You could also opt to add error bars.

Row 405: This sentence has not sense put here without any connection with your results.

Rows 518-519: Why are you reporting only the separation of blocks? I think that also plots are separated.

Row 522-523: It could be useful for readers have a picture of your experimental design, actually it is not clear if in each plot you have planted the 3 crops in strips or if each plot has 1 crop and if you planted each year the same crop in the same place or if did a crop rotation. Please integrate this information into this section to better understand your experimental design.

Roow 565: 80 L?

Rows 686-687: This introduction in my opinion is useless. Maybe you can focus on soil fertility management or the object of your research to start the conclusion.

Rows 687-689: this sentence is not clear, probably you missed a verb. Rewrite please.

Author Response

Dear Reviewer,

We would like to express our sincere appreciation and thankfulness for the great comments/suggestions made during the review of our manuscript. We have taken our time to respond as much as possible to address the concerns, and where it is not possible, we explained and hope it will meet your kind consideration without impacting the quality of the paper.

Thank you much!

Sincerely,

Lord A

Reviewer 2 Report

    • A brief summary  (one short paragraph) The aim of this paper is to study the effects of Municipal Solid Waste compost (MSW) when applied to soil, in order to investigate how it affects soil health, soil microbiome, and crop productivity and TCA cycle intermediate metabolites.  The aim of the article is clear and the focus of the study is highly interesting. The contents have novelty and are clearly presented and well organized. The research community should have interests on it. However, some improvements should be considered before publication. 
    • General concept comments
      Article:
    • The study has involved a wide range of parameters and a loto of data have been collected; it is not well clear the number of analysis performed each year of study; please provide more information about the collected data, clarifying if they come from froma year 1, year 2 etc. (in particular for data about plants productivity) , or if they have been collected at the end of the 5 years study (for data about soil). 
    • Furthermore, chemical and physical characterization of experimental soil and compost before the starting of the study are recommended. 
    • The discussion should be improved: please try to explain better the relationship between the application of MSW and the obtained results on  plants, soil and microbiome, distinguishing among the three species considered.
    • Specific comments 
    • Title is not clear and some conjunctions are missing; please review it.
    • Table S2: units of measurement are missing for the amount of elements contained into MSW
    • Line 108: The sentence is not clear: do the authors mean "soil water content"? 
    • For all the tables (n° 1, 2,3, 4, 5, 6): please provide the standard deviation values.
    • Figure 2B and Figure 3B, figure 4: please provide the y-axis name. Provide standard deviation as well.
    • Line 315: the definition of SPAD value is not coherent with  the following definitions and it is confusing. Please explain better what do you mean with SPAD and which kind of reletionship exist between SPAD value and chlorophylls contents. 
    • Line 341: the reference to Table 5 for SRL and biomass data, is wrong. They are referred to Table 6. 
    • Line 373: "Elemental accumulation in Harversted Crop". The authors present the data about elemental concentration and refers to Abbey et al. to declare the safety of the results. Please refers the obtained data also to national and international regulation about food safety.
    • Line 639: how do the authors separate the roots from soil, avoiding any root's material loss? Are the authors sure that the measured roots biomass involve the whole roots biomass grown into the soil? Please describe the method applied to isolate roots biomass. 
    • Line 657: please repeat and specify the reference used for this method 
    • It is not clear the number of replication for soil and plants chemical analysis. On lines 677-679, the authors state three replicates for soil and two replicates for plants analysis: Are these analytical replicates or sample replicates? Please specify the number of samples analyzed for plant species and each treatment with ICP-MS and  LC.MS. It is suggestable to provide a table/plot where the experimantal design is explained. 
    • Soil and plants analys have been repeated each year of study?

Author Response

(The authors gave the same response as above.)

Reviewer 3 Report

Dear Authors

The current study entitled “Application Frequency of Municipal Solid Waste Compost on Soil Physiochemical Properties and Microbiome Function, and Vegetable Crops Productivity and TCA Metabolites” is good. For a better understanding in-depth, it is a need for time to work on this topic. Furthermore, achieving potential benefits by using current technology depends on extensive research work for more exploration. Although the experiment is well organized, I suggest a rejection due to the following deficiencies.

Major Concerns

Title

  • The title doesn't indicate anything new about the present study. To grab the reader's attention, the title must accurately reflect the value addition that the writers created in their prior work. If the writers have discovered new information that hasn't been reported before, please update it in the tile to attract the attention of the reader.

Abstract

  • There is no systematic abstract. i.e., topic introduction, issue description, justification for the selection of current technology adopted in the study, knowledge gap filled, methodology in a few sentences, findings, and a conclusion.
  • Please provide the need of study in the abstract in 1-2 lines.
  • Please give a clear-cut point problem source as a problem statement that is tackled in the current study.
  • Give a logical reason for selecting the current strategy. What are the potential benefits associated with the current study?
  • No quantitative information is added in the abstract. While explaining the results, please provide some quantitative data.
  • I am unable to draw any firm conclusions from the existing research. Please provide more details especially new findings in the conclusion section.
  • Please conclude with a statement that shows a knowledge gap covered, potential beneficiaries, and specific recommendations. This statement Overall, the crops perform best in the AN-soil followed by the BI-soil is very general. It is not a conclusion.
  • Give future perspective in a single line. At least declare one best result.
  • As per standard suggestions, please avoid using title words as keywords.

Introduction

  • Also, provide a novelty statement at the end. What new things have authors done or correlated in this research compared to old ones?
  • Where is a hypothesis which is tested in the current study? I did not find any hypothesis in the current study introduction.
  • Would you please give a single line about the knowledge gap your research has covered along with the SMART (specific, measurable, achievable, realistic, and time-specific) hypothesis statement?
  • No aims of the study are provided at the end of the study. Please provide that.

Material and methods

  • It is fine.

Results

  • I request that the authors they can provide parallel plots for a better understanding of the data.
  • Also, explain the positive and negative correlation of studied attributes with applied treatments.
  • Chord diagrams can also be made to clear the percentage contribution of each studied attribute.

Discussion

  • It is ok.

Conclusion

  • The conclusion is so much descriptive. Please provide a conclusive conclusion.
  • Add the targeted beneficiary audience who will get benefit from this research.
  • Also, give clear-cut recommendations
  • Give future perspective regarding this research.

Regards

Author Response

(The authors gave the same response as above.)

Round 2

Reviewer 1 Report

I thank the authors for their work. I appreciated the changes made to the manuscript. Now it can be published. 

Reviewer 2 Report

Now I think  the manuscript has been sufficiently improved for publication on Plants.

Reviewer 3 Report

Dear Authors

I am satisfied with all the changes incorporated in the manuscript. 

Regards